# Physicochemical Characterization and Drug Release Properties of Methyl-Substituted Silica Xerogels Made Using Sol–Gel Process

**DOI:** 10.3390/ijms22179197

**Published:** 2021-08-25

**Authors:** Adél Len, Giuseppe Paladini, Loránd Románszki, Ana-Maria Putz, László Almásy, Krisztina László, Szabolcs Bálint, Andraž Krajnc, Manfred Kriechbaum, Andrei Kuncser, József Kalmár, Zoltán Dudás

**Affiliations:** 1Neutron Spectroscopy Department, Centre for Energy Research, Konkoly-Thege 29-33, 1121 Budapest, Hungary; len.adel@ek-cer.hu (A.L.); almasy.laszlo@ek-cer.hu (L.A.); 2Faculty of Engineering and Information Technology, University of Pécs, Boszorkány Str 2, 7624 Pécs, Hungary; 3Department of Mathematical and Computer Sciences, Physical Sciences and Earth Sciences, University of Messina, Viale Ferdinando Stagno D’Alcontres 31, 98166 Messina, Italy; gpaladini@unime.it; 4Functional Interfaces Research Group, Institute of Materials and Environmental Chemistry, Research Centre for Natural Sciences, Hungarian Academy of Sciences, Magyar Tudósok Körútja 2, 1117 Budapest, Hungary; romanszki.lorand@ttk.hu; 5“Coriolan Drăgulescu” Institute of Chemistry Timisoara, 24 Mihai Viteazul Ave., 300223 Timisoara, Romania; lacramaanamaria@yahoo.com; 6Department of Physical Chemistry and Materials Science, Budapest University of Technology and Economics, 1521 Budapest, Hungary; laszlo.krisztina@vbk.bme.hu; 7Semilab Semiconductor Physics Laboratory Co. Ltd., 4/A Prielle Kornelia Str., 1117 Budapest, Hungary; szabolcs.balint@semilab.hu; 8Department of Inorganic Chemistry and Technology, National Institute of Chemistry, Hajdrihova 19, 1001 Ljubljana, Slovenia; andraz.krajnc@ki.si; 9Institute of Inorganic Chemistry, Graz University of Technology, Stremayrgasse 9, 8010 Graz, Austria; manfred.kriechbaum@tugraz.at; 10National Institute of Materials Physics, 405A Atomistilor Street, 077125 Magurele, Romania; andrei.kuncser@infim.ro; 11MTA-DE ELKH Homogeneous Catalysis and Reaction Mechanisms Research Group, Department of Inorganic and Analytical Chemistry, University of Debrecen, Egyetem tér 1, 4032 Debrecen, Hungary; kalmar.jozsef@science.unideb.hu

**Keywords:** hybrid silica materials, sol–gel technique, structural characterization, nanostructure, controlled release, Captopril

## Abstract

In this work, a multi-analytical approach involving nitrogen porosimetry, small angle neutron and X-ray scattering, Fourier transform infrared (FTIR) and nuclear magnetic resonance (NMR) spectroscopies, X-ray diffraction, thermal analysis and electron microscopy was applied to organically modified silica-based xerogels obtained through the sol–gel process. Starting from a tetraethoxysilane (TEOS) precursor, methyltriethoxysilane (MTES) was added to the reaction mixture at two different pH values (2.0 and 4.5) producing hybrid xerogels with different TEOS/MTES molar ratios. Significant differences in the structure were revealed in terms of the chemical composition of the silica network, hydrophilic/hydrophobic profile, particle dimension, pore shape/size and surface characteristics. The combined use of structural characterization methods allowed us to reveal a relation between the cavity dimensions, the synthesis pH value and the grade of methyl substitution. The effect of the structural properties on the controlled Captopril release efficiency has also been tested. This knowledge facilitates tailoring the pore network for specific usage in biological/medical applications. Knowledge on structural aspects, as reported in this work, represents a key starting point for the production of high-performance silica-based hybrid materials showing enhanced efficacy compared to bare silica prepared using only TEOS.

## 1. Introduction

To meet the challenges of medical, pharmaceutical, cosmetic industry, renewable energy and environmental science, the application of interdisciplinary knowledge has become vital [1,2,3,4,5,6]. The sol–gel technique is one of the most suitable and employed techniques to develop porous structures of oxide materials for the applications mentioned [7,8,9,10,11].

The preparation of silica materials traditionally uses silicon alkoxides (tetramethoxysilane (TMOS), tetraethoxysilane (TEOS)) or sodium silicate as precursors [12,13,14]. By introducing organically modified silica precursors in the synthesis process, the resistance to mechanical stress and the hydrophobicity of the obtained silica gels can be improved. The most used substituted silica precursors for the bare silica functionalization are methyltriethoxysilane (MTES) or methyltrimethoxysilane (MTMS). The recent developments on hybrid silica materials have proven the utility and applicability of MTES and MTMS precursors in the following various fields: gas storage and separation [15,16], biocatalysis [17], corrosion protection [18], adsorption of organic solvents [19]. The final product properties are simultaneously influenced by synthesis temperature, solvent type, pH, concentration, catalyst, as well as aging and drying parameters [20,21]. The silica network formation follows several sequential pH-dependent steps including precursor hydrolysis, monomers polycondensation, solid particles creation, particle growth, particle aggregation and the strengthening of the gel network by aging and drying [22,23]. The use of acid or base catalyzed sol–gel routes modifies the primary particle formation pathway, implying different particle shapes. The hydrolysis achieves the lowest reaction rate at pH = 7. For example, at pH < 7, it is well-known that hydrolysis and condensation occur successively, the polycondensation process mechanism consists of cluster–cluster aggregation and primary particles are represented by linear or weakly branched polymer chains. However, under basic conditions (pH > 7), hydrolysis and condensation occur simultaneously, the silica polycondensation follows cluster–particle mechanism and primary silica particles look similar to highly branched clusters. Fluoride anions play an important role as sol–gel process catalysts by significantly increasing the rate of condensation reactions.

MTES has a lower affinity towards a hydrolysis reaction compared to tetrafunctional tetraalkoxy-precursors, the Si–C bond being inert to hydrolysis. The final physicochemical properties of the hybrid organic–inorganic materials depend on the extent of the hydrolysis–condensation reactions, the branching degree of the polymers and the gel homogeneity. Consequently, it is important that the methyltriethoxysilane behavior in the sol–gel environment be deeply studied.

Spectroscopic techniques such as UV–Vis [24], FTIR [25,26,27], Raman [27] and MS [28] proved to be proper and fundamental tools to study the hydrolysis/condensation reactions, essential for the silica network formation, starting from a methyl-substituted silica (co-)precursor. The influence of different solvents (methanol, ethanol, 1,4-dioxane, acetone) and synthesis temperature upon the kinetic of hydrolysis reaction was followed by FTIR [29]. As expected, higher temperatures promote the hydrolysis of the precursors. Interesting behavior of the MTES was determined in case of each parental solvent (methanol and ethanol). The reaction rate constant was five times higher in methanol than in ethanol [29]. Even though the methyl-substituted silica materials showed superhydrophobic behavior, surface hydroxyl groups were identified using FTIR [30].

In silica aerogels, 20% of the TEOS replacement with MTES increases the apparent surface area [31]. By increasing the MTES/TEOS molar ratio, the gelation time and the final product’s hydrophobicity also increase. At the same time, the particle size, density, apparent surface area and surface roughness decrease [32]. Recently, our group reported that the gradually methyl functionalized silica xerogels synthesized at pH = 3 and pH = 4 showed similar porosity and particle size alterations at the meso/micro level [33]. Methyl-substituted TEOS systems have been studied using nitrogen porosimetry [34], TGA/DTA [35], FTIR [36], NMR [37], XPS [38], XRD [39] and inelastic neutron scattering [40,41]. To the best of our knowledge, few results have been reported about the physicochemical characterization of methyl-substituted silica gels.

The presence of methyl groups in the silica gel structure was clearly distinguished in the spectrum by the presence of peaks related to the different vibrational modes of the Si–CH_3_ bond [42]. The ^29^Si-NMR spectroscopy offered complementary information about the level of the hydrolysis and condensation on the silica gels, both in the solution and solid phase [43]. X-ray photoelectron spectroscopy (XPS) was used for quantitative elemental analysis. Comparing calculated and experimental atomic ratios for the TEOS-derived gels and those synthetized from TEOS/MTES mixtures, the differences were significantly diminished for the methyl-substituted gels, meaning that the residual solvent content is lower in the methyl-substituted gels [38].

Captopril is highly water soluble, unstable in an alkaline pH and its bioavailability decreases in the presence of food [44]. To avoid these drawbacks, solid carriers can be used to improve the therapeutic efficacy and reduce the side effects of the drug [45]. A variety of carriers, such as porous metal oxides [46], micelles [47], hydrogels [48], biopolymers [49] and silica [50,51] have been employed as vehicles for Captopril delivery. In addition to these characteristics, mesoporous silica proved to be a promising drug carrier due to its excellent biocompatibility, high stability, large apparent surface area and the ease of surface functionalization.

The most innovative research works in this field focus on the design and fine tuning of these systems for different applications, especially for medical and pharmacological ones. The aim of the present study was to evaluate the influence of the pH and the effect of the introduction of organic moieties (methyl groups) on the performance of the materials. The silica xerogels were obtained using the sol–gel process and characterized using various methods in terms of composition (FTIR, XRD), texture and morphological properties (nitrogen adsorption, SAS, electron microscopy), hydrophobicity (contact angle) and thermal stability (thermal analysis). Taking into account the structural properties, selected materials were tested as controlled release carriers for Captopril. The relationship between the textural properties of the silica support and the drug loading and release efficiency was evaluated.

## 2. Results and Discussion

In this section, the results obtained on the methyl-substituted silica gels are presented. The A and B labels represent the gels obtained at pH = 2.0 and pH = 4.5, respectively. These two pH values were chosen taking into account the pH value where the point of zero charge is found and the lowest condensation rate is taking place [52,53]. The percentage (%) is the substitution grade (see detailed description in Section 3.1).

### 2.1. Gelation Time

One possible future application of the silica gels is as host materials for biomolecules, especially enzymes. In order to preserve the high activity of the immobilized biologically active guest molecules, short gelation time is one of the key factors of the encapsulation process. Gelation time is affected by all the synthesis parameters.

In the present paper, the studied factors that influence the gelation time were the precursors’ molar ratio and type, and the pH. Rios and coworkers used the same precursor ratio, but the gelation time was much longer than in the present case [32]. Figure 1 reports the evolution of the gelation time of all the investigated silica samples using different amount of MTES and catalysts.

The gelation time exponentially increased with the MTES molar ratio, and a shorter gelation time was observed for the gels prepared at a less acidic pH.

### 2.2. Low-Temperature Nitrogen Adsorption/Desorption

In order to characterize the porous texture of the synthetized gels, low-temperature nitrogen adsorption has also been applied (see adsorption/desorption isotherms in Figure 2a and Appendix A).

For series B, a transition caused by the increasing amount of MTES was observed, the hysteresis loop changed from Type H3 to H2a. Type H3 hysteresis was found for the non-substituted sample, characteristic for large cone-like pores [54,55]. A small addition of MTES (B5) caused the change of the H3 loop to H2a-type hysteresis. Starting from sample B5, a larger amount of mesopores developed, which had gradually been changing up to the B60 sample: the size of the mesopores narrowed and their proportion reduced. The volume of the mesopores decreased, and the volume of micropores increased. The highest methyl substituted samples from the B series showed the H4 hysteresis type.

Almost all the methyl-substituted gels showed a higher surface area than the samples without substitution (Appendix A). For series B, an inverse parabolic evolution of the apparent surface area could be observed, with a maximum at 40% MTES content (Figure 2b).

The surface area of 1115 m^2^/g was comparable to the values obtained for silica aerogels or ordered mesoporous silica [56,57]. For series A, a completely different evolution was observed: except the highest MTES substitution, an almost constant surface area was obtained. The majority of the samples possessed a surface area of ~600 m^2^/g. The average pore size diameter is presented in Figure 3. Cylindrical pore geometry was assumed. For series A, the pores were smaller than 5 nm in diameter; however, for series B, the pores size decreased with the increasing MTES percentage. Very high pore diameter values (above 10 nm) were obtained for the lowest MTES concentrations (0%, 5%) of series B (pH 4.5); however, most samples showed diameters slightly above 5 nm, and only the two highest methyl concentrations caused a shift of the porosity towards the microporous domain.

A similar evolution could be observed for the volume of the microspores with an increasing MTES concentration. The maximum value of the microporosity was found for the samples with 40, 50 and 60% MTES. For the majority of the series A samples, at least half of the pore volume was found to be in the microporous domain.

### 2.3. Small Angle Scattering (SAS)

Small angle neutron (SANS) and X-ray (SAXS) scattering studies have been performed on all the methyl-substituted hybrid silica xerogels. The data have been analyzed by applying the combined model of the following Guinier and Porod approximations, as introduced by Hammouda [58] (Equations (1) and (2)):(1)$$I\left(Q\right)=G\;\exp\left[-Q^{2}{R_{g}}^{2}/3\right];Q\leq Q_{1}$$
(2)$$I\left(Q\right)=D/Q^{p}\;;Q\geq Q_{1}$$
where *G* and *D* are the Gunier and Porod scale factors, *Q* is the scattering vector, *I*(*Q*) is the scattered intensity, *R*_g_ is the radius of gyration and *p* is the power law exponent. *Q*_1_ represents the scattering vector value, where the intensity values and the slopes of the Guinier and Porod terms are equal.

At the smallest angles, the characteristic size of the particles and/or pores dominates and is described by the Guinier approximation (1), while the high angle part of the scattering curve shows a power law behavior and describes the surface roughness of the particles (2).

The SANS and SAXS curves are shown in the Appendix A

Similar to the results obtained using low-temperature nitrogen porosimetry and electron microscopy, the SANS and SAXS measurements confirm that in the nanoscale region, the series A and B, prepared at a different pH, are different in their nanostructure (Figure 3, Appendix A).

Series A shows a maximum diameter of the scattering objects at 50% MTES substitution; however, the same peak for series B is much less pronounced. Since in low-temperature nitrogen porosimetry could not be observed any increase in pore size by increasing the MTES content, the existence of an increased measured size in SAS can be explained by the stronger agglomeration of the small primary particles, resulting in an average cluster size of 25–30 nm for series A and 10–12 nm for series B. SAS confirms the slightly larger (series A) and larger (series B) pore sizes in the case of low MTES substituted samples. For both series, at a low MTES substitution, exponent *p* varies between three and four, characteristic of fractal-like surfaces.

For MTES content above 30%, exponent *p* decreases below three, which would indicate surface fractal properties. However, at this composition range, the Guinier type scattering becomes weaker, and interferes on a wider *Q* range with the exponential type scattering that arises not only from the pores’ surface but also from the primary and aggregated particles. Therefore, those *p* values can be regarded as only apparent fit parameters that ensure the smoothness of the high *Q* part of the Guinier approximation.

SAS, N_2_ adsorption and electron microscopy show that the gradual increase in the number of –CH_3_ groups changes the pore and particle structure. It can be assumed that the methyl groups are found at the surface of the silica particles. The pores’ dimension is decreasing with the increase in the MTES/TEOS molar ratio, especially for pH = 4.5.

### 2.4. Electron Microscopy

#### 2.4.1. Transmission Electron Microscopy (TEM)

TEM images of the prepared silica materials are shown in Figure 4. All the samples were composed of plate-like silica particles. TEM confirmed the previously (SAS and BET) presented results that important morphological changes in the silica hybrids occurred with an increasing MTES concentration: the porosity of samples decreased, and the morphology changed from an amorphous unordered to a compact structure with square-like particles without any visible porosity. This tendency was valid for both series. The appearance of ordered domains was also supported by the XRD results. At the highest magnification for the A0, A5 and B5 samples, the primary particles’ sizes could be seen. Their size varied between 10–20 nm. In agreement with the SAS results, in the case of 40 and 50% methyl-substituted samples, the primary particles were larger for series A than for series B.

#### 2.4.2. Scanning Electron Microscopy (SEM)

SEM images (Figure 5) show the surface topology of silica xerogels. With increasing MTES content, the particles became more compact and edged. The plate-like structure observed in TEM was present at micrometer sizes, as well.

### 2.5. FTIR-ATR Spectroscopy

FTIR-ATR spectra were collected for all the xerogels. The spectra for series A, in the 400–4000 cm^−1^ wavenumber range, are displayed in Figure 6.

Vibrational modes of the silica network, more prominent in the case of TEOS (0% MTES), were detected at ~455, ~800 and ~1065 cm^−1^. The low frequency contributions (~455 and ~800 cm^−1^) are bending modes of the Si–O bonds, whereas the high-intensity band at ~1065 cm^−1^ is attributed to the Si–O–Si stretching mode of the SiO_2_ skeletal structure. By increasing the amount of MTES in the reaction mixture, significant differences were observed when compared to the spectrum of the pure TEOS xerogel. An enhancement of the intensities associated to the characteristic Si–CH_3_ bands (~1275 and ~2975 cm^−1^) was observed, which was assigned to the symmetric deformation of Si–CH_3_ and the non-hydrolyzed ethoxy groups, respectively. At the same time, bands belonging to the ν_as_ Si–O–Si (1030–1080 cm^−1^)_,_ δ_s_ C–H (1272–1279 cm^−1^) and ν_s_ C–H (2972–2980 cm^−1^) shifted to lower wavenumber values. In Figure 7, the evolution of the Si–O–Si asymmetrical and the C–H symmetrical stretching vibration bands as a function of the MTES substitution degree are shown.

A slight contribution of the H–O–H bending mode of water molecules at 1631 cm^−1^ was observed (see Figure 6). It is worth mentioning the shift towards lower values exhibited by the 3000–4000 cm^−1^ band, ascribed to the fundamental OH stretching vibration of different hydroxyl groups, suggesting an increasing hydrophobicity of the silica gels with an increasing methyl content. The band intensity decrease was the most pronounced in the case of the samples without methyl groups and with a low concentration of MTES. Accordingly, a reduction in the band intensity as the MTES amount increased is an experimental indication of the increased hydrophobicity of the investigated materials. The same tendency can be observed for the symmetric vibration of the Si-OH bond, evidentiated by a shoulder at approximatively 950 cm^−1^.

The maximum was moving to lower values with the increasing concentration of the MTES (Figure 7). This shift was explained with the increasing distance between the silica planes with the increasing number of the methyl groups, which was attributed to the ratio modification of LO_4_-TO_4_ vibrational modes (ν_as_(Si–O–Si)TO at ~1075 cm^−1^ and ν_as_(Si–O–Si)LO at ~1180 cm^−1^) and caused a decrease in porosity or an increase in the system density, as expected, in agreement with the SAS data, electron microscopy and DFT modelling of low-temperature nitrogen porosimetry data.

### 2.6. X-ray Diffraction

Diffractograms in the range of 2*θ* = 5–35° were recorded for the ground/crushed xerogels, prepared with a different amount of MTES. One specific maximum was observed in the 2*θ* = 22–23° region (Figure 8). The maximum moved to lower values with the increasing MTES concentration (Appendix A), indicating that with the increasing quantity of the methyl groups, the spacing between silicon atoms connected by an oxygen bridge has also increased. The higher methyl content (>60%) caused the appearance of a second maximum centered around 9°, explained by the appearance of ordered domains and the presence of four-fold siloxane rings. These results are in good agreement with the results obtained using FTIR spectroscopy and electron microscopy.

### 2.7. ^29^Si MAS NMR

The solid-state ^29^Si NMR spectra (Figure 9) of the TEOS and MTES/TEOS derived materials showed a gradual increase in peaks T^3^ and T^2^ with the increase in the MTES content. The intensities of the Q^3^ and Q^4^ peaks showed a reverse trend. The Q^4^ position shifted to lower values, while the T^3^ and T^2^ position shifted to higher values with an increasing MTES concentration.

The strong Q^4^ and T^3^ signals revealed a high condensation extent of TEOS and MTES. The quantitative analysis of the ^29^Si MAS NMR spectra is summarized in Appendix A. For samples derived from TEOS, 71% of the silica network was fully condensed (Q^4^ species). The incorporation of a small amount of trifunctional (T^i^) units into the neat silica network caused a decrease in the condensation rate of the tetrafunctional structural units. This indicates the initiation of disordering in the siloxane network due to the co-condensation. Therefore, a small amount of methyl substituents acts as defect-forming agent in the network. Due to steric effects, not all the silanol groups are able to condense, and remain free (confirmed also by ^1^H MAS NMR). However, with an increasing amount of T^i^ units in the network, the condensation rate of Q^i^ units subsequently increased as a result of the higher reactivity and free volume provided by the methylsiloxane units. In this way, highly condensed copolymer siloxane networks were formed. In case of the samples with 100, 95, 90 and 60% TEOS, the tetrafunctional structural units were transformed to Q^4^ species in 70% extent, while in the case of the 40 and 20% samples, around 90% of the sites were transformed to Q^4^.

Table 1 compares the ^29^Si MAS NMR results of samples synthetized at pH = 2 (series A) and pH = 4.5 (series B). A more acidic pH promoted a faster hydrolysis and a slower condensation of the precursors, while at a less acidic pH, the hydrolysis and condensation reactions were increasing or decreasing in parallel. In addition, the CH_3_–Si containing sites had lower reactivity. All these factors together favored the formation of Q^4^ species at pH = 4.5, while at pH = 2, a larger amount of T^3^ species were formed. For both pH values, the condensation level increased with the increasing quantity of the MTES ratio in the precursor mixture.

There are the following two possible explanations for the aforementioned shift of both contributions: either (i) the Q^4^ and Q^3^ ratio is increasing, or (ii) the diminishing of the bond strength associated to the Si–O–Si bonds may lead to longer Si–O–Si distances. The second explanation is in good agreement with the XRD measurements, where the increasing quantity of the methyl groups led to larger silica plane distances.

For the samples synthesized with 0, 5 and 10% MTES, the high degree of the three-dimensional (3D) cross-linking gel network was observed (Q^4^ + Q^3^ + T^3^ = 97–99%), while for the B40, A40, B60, A80 or B80 samples, the 3D cross-linking degree was smaller (80–91%). The sum of the Q^2^ and T^2^ signals suggested a linear silica (2D) network (Table 1 and Appendix A). The proportion of the 2D silica network increased gradually from 1 to 19% with the amount of methyl functionalization. The hydrophobicity of the samples could be linked indirectly to the amount of hydroxyl groups. The silica sites with hydroxyl groups were represented by the Q^3^, Q^2^ and T^2^ species. The gradual increase in the methyl substitution caused a decrease in the sum of Q^3^ + Q^2^ + T^2^, which means an increasing hydrophobicity, as supported by the thermal analysis, FTIR spectra and confirmed also by the contact angle measurements. The sum of the T^x^ species (Table 1 and Appendix A) shows the degree of methylation of every sample.

### 2.8. Thermal Analysis

The thermal behavior of silica and methyl-substituted silica xerogels was studied using thermal analysis. All the synthetized samples showed a typical decomposition profile with three distinctive mass loss steps (Appendix A). The initial mass loss (up to 250 °C) was attributed to the removal of water. The percent of the mass loss (Table 2) depended on the chemical composition of the xerogels. In the temperature range from 250 to 650 °C, a main mass decrease (up to 12%) with expressed exothermal effect (DTA curve), due the thermal decomposition of methyl groups, was observed. At temperatures above 650 °C, the TG curves showed only a slight mass loss (up to 1.2%), corresponding to water loss due to the condensation of the silanol groups to form siloxane bonds. The total mass loss was about 3% for the sample made from TEOS and 12–13% for the samples with an MTES content. It is interesting to observe that all the methyl-substituted samples showed around 12% total mass loss, which is caused by the thermal decomposition of the methyl groups.

### 2.9. Contact Angle

The representative contact angle (CA) images are presented in the Appendix A. The CA data for all types of samples are presented as box plots in function of the methyl content (Figure 10). For ease of comparison, the data were grouped in pairs by the following two properties: second step catalysis type (series A vs. series B) and state of the powder (as-received vs. ground). Due to the difficulties of measuring caused by the powder state of the samples, the range of CAs of the samples with identical methyl content was high, typically 10–20°.

The CA dependence on the methyl content and catalyst type was similar to the trends observed for the vinylated silica xerogels [59].

The CAs follow a generally increasing trend with the increase in the methyl content. The median CAs range from 13.5 to 153.5°. The same CA dependency versus methyl content was also observed for other methylated silica xerogels [19,60,61] and could be explained along the Cassie–Baxter theory [62]. According to this theory, the surface of the studied xerogels is composed of “bare” silica with a low intrinsic CA, *θ*_silica_ and methyl groups with a high intrinsic CA, *θ*_methyl_.

The CA of the xerogel, *θ*, depends on the CA of these two surface components as follows:(3)$$\cos\theta =\left(1-f_{\mathrm{methyl}}\right)\cos\theta_{\mathrm{silica}}+f_{\mathrm{methyl}}\cos\theta_{\mathrm{methyl}},$$
where *f*_methyl_ denotes the surface fraction of the methyl groups. With an increasing MTES mol%, this surface fraction should necessarily increase, therefore contributing to the increase in the total CA.

Series B reached a higher maximum CA than series A. As evidenced by the low-temperature nitrogen sorption (Figure 2b), the apparent surface area of series B was higher than that of series A. According to the Wenzel model [63], the relative surface area affects the measured CA as follows:(4)$$\cos\theta_{W}=r_{W}\cos\theta_{Y},$$
where *θ*_W_ stands for the real (“Wenzel”) CA of the rough surface, *θ*_Y_ is the ideal equilibrium (“Young”) CA of the same flat surface and *r*_W_ is a dimensionless roughness factor, the ratio of the real surface area to its geometrical projection. According to this relation, for hydrophilic surfaces (*θ*_Y_ < 90°), a higher than geometrical surface area leads to a lower real CA, whereas increasing the real surface area of hydrophobic surfaces (*θ*_Y_ ˃ 90°) would result in even higher CAs. Along this model, the higher maximum CA of series B can be explained by the higher surface area.

Series B reached the maximum CA at a lower methyl content (as-received samples: 50%; ground samples: 60%) than series A (as-received samples: 70%; ground samples: 80%).

For both series A and B, the as-received and ground samples exhibited identical contact angles at a low and high methyl content but deviated at an intermediate (20–50%) methyl content.

### 2.10. Captopril Release

A Captopril amount (217.3 mg of Captopril) equal to the necessary dosage for an adult in 24 h was loaded into the selected silica sample. Loading was realized either by adsorption in the as-obtained material or entrapped in situ during the sol–gel synthesis [64]. The aim was to evaluate the effect of the structural properties of the silica carrier on the controlled release of Captopril. The Captopril release kinetics were correlated to the methyl-substituted silica morphologies, textural properties and hydrophilic/hydrophobic profiles. For the samples loaded in situ during the sol–gel process, three methyl-substituted silica carriers had been selected based on their different hydrophobic/hydrophilic profiles and the controlled release of Captopril was evaluated in two media. The release media were chosen to simulate the intestinal fluid (pH = 7.4) and the acidic media of the stomach (pH = 1.2).

It was shown in previous studies that the release of Captopril from MCM-41 carriers in simulated intestine (pH = 6.8) and gastric fluids (pH = 1.2) starts with a burst phase in which ca. 60% of the drug dissolves. This is followed by a slow retarded phase reaching ca. 90% cumulative release. The overall rate is significantly slower in simulated intestine fluid due to its near neutral pH. Under these conditions, Captopril becomes deprotonated, which causes its strong interaction with silica that, in turn, limits the release rate [65].

#### 2.10.1. In Vitro Drug Release from the Samples Loaded by Adsorption

For the adsorption-loaded B5 and B40, the in vitro drug release was tested at room temperature in a hydrochloric acid solution of pH = 1.2. The amount of released Captopril was determined using UV–Vis spectrophotometry using an external calibration curve. For the B5 sample, only 5% of the Captopril was released after 5 h. For the B40 sample, 12% of the Captopril was released after 7.6 h. This 12% of release was maintained after 24 h and increased only to 15% after 48 h (see Figure 11).

The explanation for the low cumulative drug release could be the strong interaction of the drug with the backbone of the silica carriers. In this case, there is a high probability that the different forms of Captopril, i.e., bound to the carrier and dissolved in the medium, are in a dynamic equilibrium with each other. The position of the dynamic equilibrium, and thus, the cumulative drug release, depends on the strength of the specific interactions of the Captopril molecules and the different silicas [66,67]. In a general case of a hydrophilic purely silica matrix, the interaction between the drug and the carrier backbone takes place via hydrogen bonds with the silanol groups of silica. Considering that Captopril is a polar and hydrophilic molecule, it resulted naturally that its interaction with the B5 silica was stronger than it was with the hydrophobic B40 silica. Therefore, the B5 carrier extensively limited the cumulative release of Captopril.

In general, the morphological characteristics of the carriers and the mode of drug loading have a strong effect on the release kinetics. In the case of the B5 and B40 samples loaded by adsorption, most of the Captopril was bound on the surface and on the accessible pores of the carriers, which resulted in the very fast establishment of the subsequent release equilibrium. In other words, the amount of Captopril that could desorb from the carrier—taking the strong interaction of the drug and the carrier into account—was released in a burst-like process. It is important to highlight that the very fast nature of the release and the low cumulative release are not in contradiction, because of the dynamic equilibrium nature of the process [68].

#### 2.10.2. In Vitro Drug Release from the Samples Loaded In Situ during Sol–Gel Synthesis

Three different Captopril concentrations were loaded in situ in three silica matrices. For these three samples, the in vitro drug release was tested (Figure 12 and Table 3), in both hydrochloric acid solution (of pH = 1.2) and in TRIS buffer solution (of pH = 7.4). Most of the drug was released in the first five minutes, and in the next 48 h, only an additionally small amount of drug was released. The maximum release of the Captopril (ca. 70%) was obtained in the simulated stomach acidic media using the silica support with the highest apparent surface area B40.

Similar to the samples loaded with adsorption, the amount of the released drug from the in situ loaded carriers depends on the strength of the interaction of the drug with the different carrier matrices. Furthermore, these interactions are highly dependent on the pH of the medium, which is represented by the release curves. In HCl at pH = 1.2, the silica carriers were expected to have a positive surface charge, and the Captopril molecules to be protonated. In the TRIS buffer at pH = 7.4, the silicas were expected to have a negative surface charge and the drug molecules to be deprotonated [69,70]. Therefore, in TRIS buffer, by the mediation of cations, Coulomb interactions and ion pair formation were expected to result in strong interactions between the carrier matrices and the drug molecules, which resulted in a limited cumulative drug release. According to the same considerations as detailed in the previous section, the low cumulative drug release was realized in a burst-like process due to the fast establishment of the dynamic equilibrium between the dissolved and the bound forms of the drug.

Interestingly, at pH = 1.2, the B5 sample showed temporal retention towards Captopril. However, this was attributed to the mode of loading the silica carrier and not the conditions of release. First, the B5 silica was hydrophilic and strongly bonded the Captopril, and second, the in situ process enabled the loading of the inner pores of the matrix, where the mass transport of the drug towards the release medium was limited. These combined effects resulted in the retention of Captopril from B5 compared to B40 (and A40) in HCl, and even compared to the adsorption-loaded B5 in HCl.

In general, the mechanism of burst release is the prompt dissolution of the drug from the surface and the open and accessible pores of the carrier particles. Such drug release is not governed by swelling, erosion or limited mass transport (hindered diffusion), and therefore, the mathematical models developed for describing these processes are not adequate for the carriers described in the present paper.

The mathematical modeling of the kinetics of drug release was informative only in the case of the A40 silica carrier in HCl medium (Appendix A). In this case, the kinetic behavior of the system could be adequately described with a simple first order kinetic model, which is in good agreement with the burst-like character of this release process [68,71]. (The details are given in the Appendix A) These kinds of matrices are suitable for the treatment of acute infections or inflammations, when an immediate high dose of drugs is required.

### 2.11. Possible Application of the Carriers

Carriers such as these xerogels are very useful for solubilizing drugs of limited hydrophilicity, and delivering them, first in a burst-like manner reaching a high concentration, and later maintaining it from the reservoir of the carrier. As the drug is absorbed in the body, the dynamic equilibrium nature of the release process replenishes it and helps to maintain a constant concentration. This is very useful, e.g., in the oral administration of drugs for the treatment of pain, acute infections or inflammations. Drugs that could benefit from loading into the studied xerogels are nonsteroidal anti-inflammatory drugs (e.g., Ibuprofen, Ketoprofen) [72,73]. Higher loading and cumulative release are expected from these drugs due to their partially hydrophobic nature. Their controlled solubilization by the studied xerogels could be advantageous for their therapeutic action.

### 2.12. Biocompatibility

It is very important to highlight that the silica-based carriers described in the present paper are porous nanostructured carriers, but not nanoparticles. Therefore, nanoparticle-associated toxicity is excluded. The material of the carriers is porous nanostructured and surface modified silica. These materials have been described several times in the literature to be non-toxic and biocompatible [74,75].

## 3. Materials and Methods

### 3.1. Synthesis

Synthesis of alkyl-substituted silica xerogels has been performed using sol–gel technique, using MTES and TEOS as precursors. The methyl-free silica gels were prepared using a solution made from TEOS, EtOH, H_2_O and HCl in molar ratio of 1/8/6/0.012. Two hydrochloric acid quantities were used; the molar ratio of them was 150/1 (pH = 2—series A/pH = 4.5—series B). Modification of silica sol was carried out by replacement of different percent of TEOS (5–80%) with MTES. These samples were labelled as XN, with X = A or B and N = substitution percentage (%). The gelation of the silica sols was obtained after addition of NaF catalyst, the precursor mixture/NaF molar ratio was settled as 50/1. The gels were kept for 24 h at room temperature for ageing and dried successively for 10 h at 40, 60 and 105 °C. All the agents were procured from Merck KGaA, Darmstadt, Germany.

### 3.2. Methods

The nitrogen adsorption–desorption measurements were performed at the temperature of the liquid nitrogen using a Quantachrome Nova 2000-e instrument (Boynton Beach, Florida, United States) to obtain the textural properties of the materials. The weight of the samples (50–80 mg) was measured exactly after pretreatment at 110 °C for 24 h. The apparent surface area *S_BET_* was determined using the Brunauer–Emmett–Teller (BET) model. A pore volume *V_tot_* was estimated from the amount of vapor adsorbed at *p*/*p*_0_ = 0.98, assuming that the adsorbed gas was present as liquid N_2_. The Dubinin–Radushkevich (DR) plot was used to calculate the micropore volume *V_micro_*. The average pore diameter was calculated with the aid of DFT model, using an equilibrium model for silica

SANS measurements were performed on the Yellow Submarine instrument at the Budapest Neutron Centre (Custom Large Scale Facility, Budapest, Hungary) [76,77]. The scattered neutrons were detected using a BF_3_ gas filled position sensitive detector with 64 × 64 cm^2^ active area and 1 cm^2^ pixel size. Using different wavelength and sample–detector distances, a range of the scattering vector modulus, *Q*, from 0.007 to 0.4 Å^−1^ has been covered (*Q* = (4π/*λ*) sin (*θ*/2), where *θ* is the scattering angle and *λ* is the wavelength of the neutrons). The measurements were carried out at room temperature.

Small angle X-ray scattering measurements were performed with a high-flux SAXSess camera by Anton Paar, Graz, Austria (for details see Appendix A).

For electron microscopy investigations, JEOL ARM200F Analytical Transmission Electron Microscope (JEOL Ltd., Tokio, Japan) and TESCAN Lyra3 XMU Scanning Electron Microscope (Tescan Orsay Holding, a.s., Brno, Czech Republic) (dual SEM-FIB instrument, equipped with FEG) have been used.

FTIR-ATR measurements were recorded in the 400–4000 cm^−1^ wavenumber range using a Bruker Tensor 37 spectrometer, Bruker Optik GmbH, Ettlingen, Germany, equipped with an overhead attenuated total reflectance (ATR). Samples were placed in contact with the surface of the ATR crystal contained in the Golden Gate diamond ATR system, based on the Attenuated Total Reflectance (ATR) technique. A resolution of 4 cm^−1^ was used, by adding 256 repetitive scans, in order to achieve highly reproducible and high-quality spectra with a good signal-to-noise ratio. A band decomposition procedure was applied on the experimental data in the 900–1300 cm^−1^ region. Accordingly, the sub-band positions were first evaluated by second-derivative computations, then Gaussian curves were adopted with all parameters free to change upon iteration until convergence was reached (*r*^2^ ≈ 0.9999).

XRD patterns were measured on a Philips X’Pert MPD diffractometer (Philips Research, Eindhoven, The Netherlands) equipped with a graphite monochromator and a scintillation counter, and having Bragg–Brentano parafocusing arrangement, using Cu Ka radiation (λ = 1.5418 Å) in theta–theta scanning mode. The samples were mildly pressed in a standard powder sample holder.

The thermogravimetric analyses were performed in air from 25 to 800 °C with a heating rate of 10 °C/min with Mettler TGA/SDTA 851/LF/1100 (TGA, Mettler Toledo, Columbus, OH, USA) thermo analyzer system.

For contact angle measurements, powder samples (both as-received and ground in a mortar) were deposited as a thin layer on double-sided adhesive tape stripes mounted on microscope slides placed on the stage of a home-built contact angle goniometer. Ultrapure water (MilliQ, *ρ* = 18.2 MΩ∙cm) was used as measuring liquid. Droplets of 2–10 µL were disposed from a 25-microliter microsyringe (Hamilton) in multiple steps. Still images and videos were recorded during the disposal of the droplets. Left and right contact angles have been measured. For each sample type, 2 to 22 droplets have been deposited, thus finally obtaining 2 to 22 values of contact angles.

### 3.3. Controlled Release of Captopril and Drug Loading Procedures

#### 3.3.1. Chemicals

Pure Captopril ((2*S*)-1-[(2*S*)-2-methyl-3-sulfanylpropanoyl]pyrrolidine-2-carboxylic acid, provided by a p*harmaceutical company free of charge*); Tris(hydroxymethyl)aminomethane (Merck KGaA, Darmstadt, Germany); Hydrochloric acid 37% (SC Silal Trading SRL, Bucuresti, Romania); NaCl (SC Chimopar Trading SRL, Bucuresti, Romania).

#### 3.3.2. Captopril Loaded by Adsorption

The following two silica samples were used for loading: B5 and B40. A classic procedure for loading was used by soaking 0.2 g of silica material with 10 mL of 0.1 M Captopril solution (containing 217.3 mg of Captopril prepared in NaCl 0.9%, with continuous stirring at room temperature. After 24 h, the material (the drug loaded mesoporous silica) was filtered and let dry at room temperature for another day. The filtrates have been tested spectrophotometrically for any free drug remains.

#### 3.3.3. Captopril Loaded In Situ during the Material Synthesis

For the in situ loading, the synthesis procedure described in Section 3.1 was used, the Captopril (1086.5 mg captopril/g of dry silica support) was added before the addition of the second catalysts (NaF or NH_4_F).

#### 3.3.4. In Vitro Drug Release Procedures

Drug release was tested in vitro at room temperature for several hours in hydrochloric acid solution of pH = 1.2 (in order to simulate the gastric pH condition) and in TRIS buffer solution of pH = 7.4 (in order to mimic the pH of intestinal fluid). The experiment was performed by soaking the obtained drug loaded mesoporous silica in 200 mL of buffer solution, under stirring, at room temperature to mimic the peristaltic motility motion from human organism [78]. At regular time intervals (5 min in the first hour; 15 min in the second hour and 1 h in the last 3 h), 3-milliliter samples were removed for analysis and replaced with 3 mL of fresh buffer solution. The 3-milliliter samples (filtered each time before analyses), being too concentrated in Captopril, were diluted each time in order to be measurable by spectrophotometer.

Solutions preparation: the 0.1 M Captopril solution was prepared in 0.9% NaCl. The hydrochloric acidic solution of pH = 1.2 was prepared from 37% hydrochloric acid. The TRIS buffer solution preparation: 100 mL of TRIS Buffer solution of pH 7.5 was prepared by dissolving 13 g of TRIS in 60 mL of distilled water. Next, the pH was adjusted to 7.5 by adding 20 mL of HCl solution (5 M). After that, the final volume of 100 mL was obtained by adding distilled water.

Calibration curve for Captopril: different calibration curves for Captopril were set. The 0.1 M Captopril stock solution was prepared in NaCl 0.9% solution in order to reach a solution with a measurable concentration at 209 nm. Next, successive dilutions of the stock solution were performed in NaCl 0.9% and in the different release media of HCl and TRIS buffer.

## 4. Conclusions

This study presented a simple approach to modify the physicochemical characteristics of the methyl substituted silica gels by varying the precursors molar ratios and the pH. The effect of the structural properties of the silica support on the controlled Captopril release efficiency was evaluated.

By a gradual increase in the MTES content in the samples, the skeletal SiO_2_ network has changed, causing changes at the nano and micro levels as well. By only varying the reaction pH and the precursors molar ratio, two very different series of samples were obtained, especially in terms of morphological and textural properties. The more acidic synthesis pH (2.5) provided samples with an almost constant apparent surface area in an increasing hydrophobic environment; an apparent surface area independent of the methyl substitution degree of ~600 m^2^/g was observed with characteristic pore size in the microporous domain. The higher synthesis pH (4.5) produced a higher apparent surface area; in this series, a maximum of 1115 m^2^/g for sample at 40% MTES substitution has been achieved.

For both series, SANS and SAXS data, electron microscopy and DFT calculations revealed that by increasing the methyl functionalization, a decrease in the porosity and increase in the structure compaction was achieved.

With the increase in the methyl content from 0 to 80%, the hydrophilicity of the silica profile changed to become less hydrophilic, as shown by the contact angle results and also supported by the FTIR and ^29^Si-MAS-NMR results.

According to the results obtained from the physicochemical characterization methods, the methyl group from the functionalized precursor determined the crosslinking degree with the TEOS-derived oligomers by blocking one site in every MTES molecule. The hydrophobic character of the methyl groups slowed down the hydrolysis, causing a longer gelation time for every methyl containing silica sol.

In the drug loading and release experiments, for the Captopril loaded by post-synthesis adsorption, 12% of the loaded amount has been released after 8 h in stomach acidic media, while for loading during the sol–gel synthesis, ca. 70% of the drug was released. In general, the specific drug loading capacity was determined by the apparent surface area, and the highest release was obtained for the sample B40, made with a 40:60 MTES:TEOS precursor ratio.

## Figures and Tables

**Figure 1 ijms-22-09197-f001:**
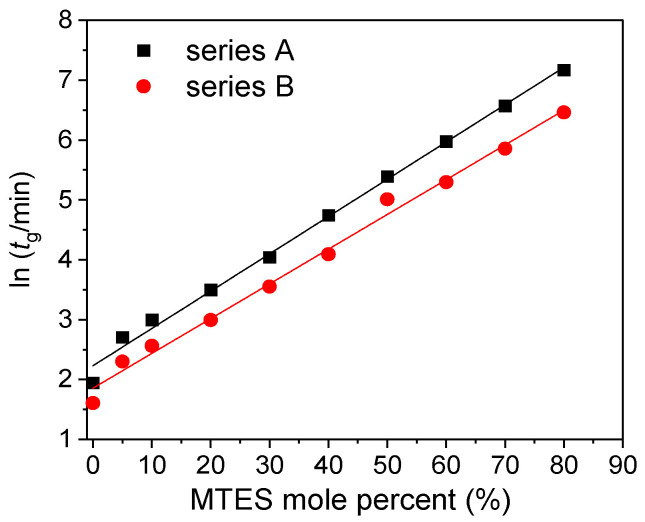
Gelation time (*t*_g_) as a function of the MTES/TEOS molar ratio at two catalyst concentrations (series A and series B labels represent the pH = 2.0 and pH = 4.5).

**Figure 2 ijms-22-09197-f002:**
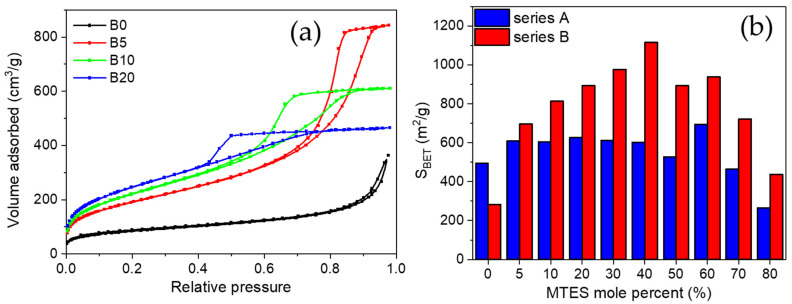
(**a**) Evolution of low-temperature nitrogen adsorption/desorption isotherms with MTES content. (**b**) Comparison between the apparent surface area evolution with the MTES mole percent and the acid catalysts content (series A and series B labels represent the pH = 2.0 and pH = 4.5).

**Figure 3 ijms-22-09197-f003:**
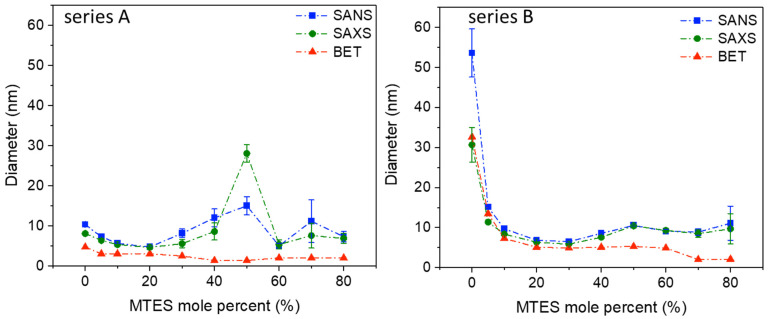
Diameter of pores from low-temperature nitrogen porosimetry DFT calculation (red), and sizes of the scattering objects—pores and particles together—from SAS (SANS—blue, SAXS—green) (series A and series B labels represent the pH = 2.0 and pH = 4.5).

**Figure 4 ijms-22-09197-f004:**
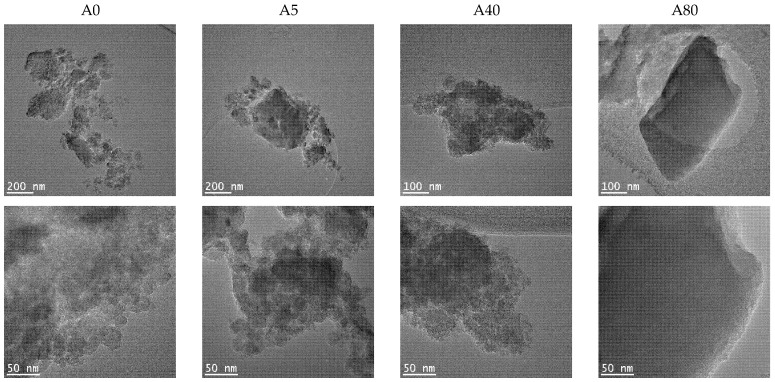
TEM images of the hybrid silica xerogels, series A and B, obtained with different methyl substituent content (series A and series B labels represent the pH = 2.0 and pH = 4.5).

**Figure 5 ijms-22-09197-f005:**
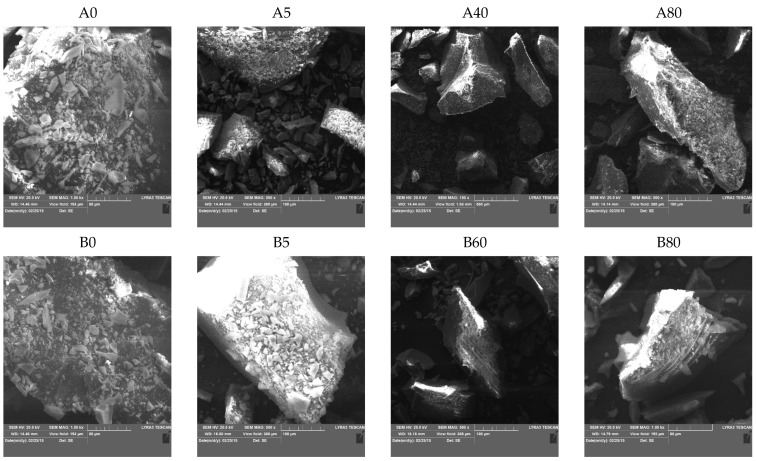
SEM micrographs of the hybrid silica xerogels, series A and B, obtained with different methyl substituent content (series A and series B labels represent the pH = 2.0 and pH = 4.5).

**Figure 6 ijms-22-09197-f006:**
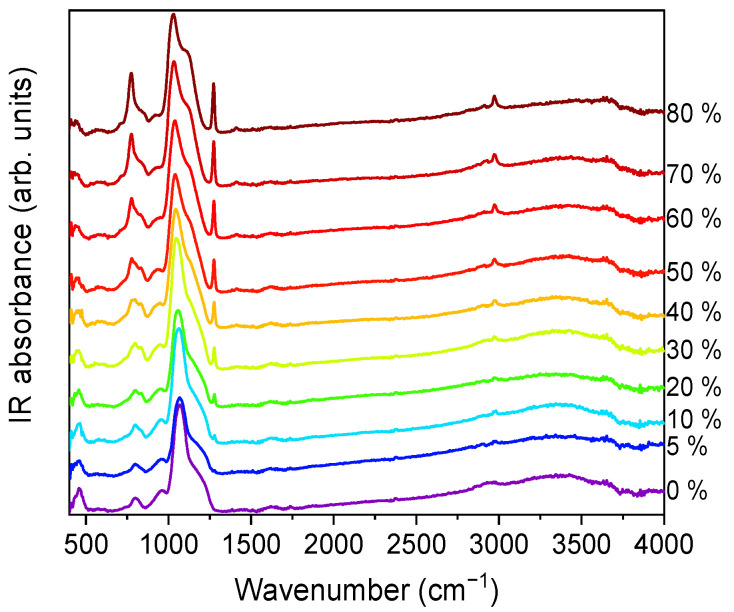
FTIR spectra of series A at different methyl concentrations. All spectra have been vertically shifted (series A represents the pH = 2.0).

**Figure 7 ijms-22-09197-f007:**
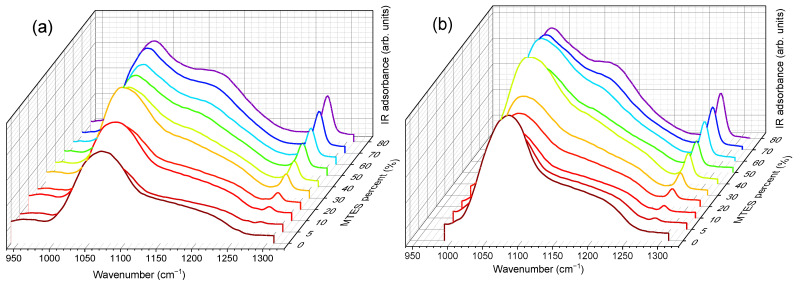
Magnification of the 950–1300 cm^−1^ wavenumber region, for (**a**) series A and (**b**) series B, where the Si–O–Si asymmetric and δ_s_ C–H vibrations are measured. The down shift of both maxima with the increase in the amount of –CH_3_ can be observed(series A and series B labels represent the pH = 2.0 and pH = 4.5).

**Figure 8 ijms-22-09197-f008:**
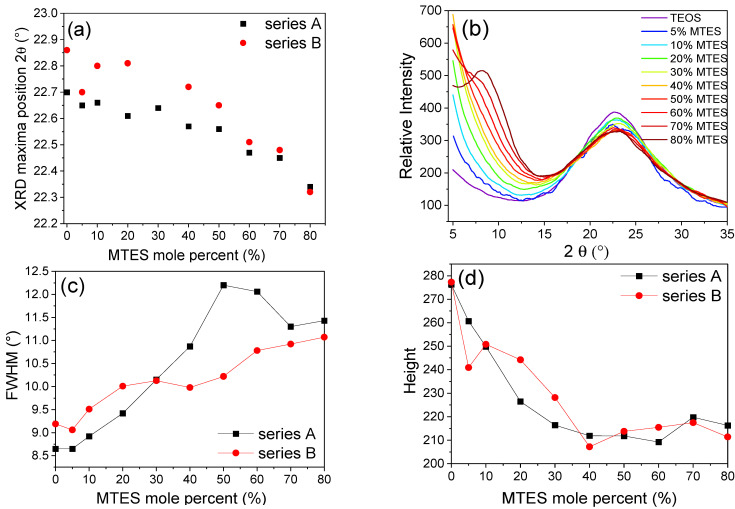
(**a**) XRD maxima position shift, (**b**) plots of the relative intensities as a function of MTES substitution, (**c**) FWHM and (**d**) peak height changes with MTES content for series A and B, respectively (series A and series B labels represent the pH = 2.0 and pH = 4.5).

**Figure 9 ijms-22-09197-f009:**
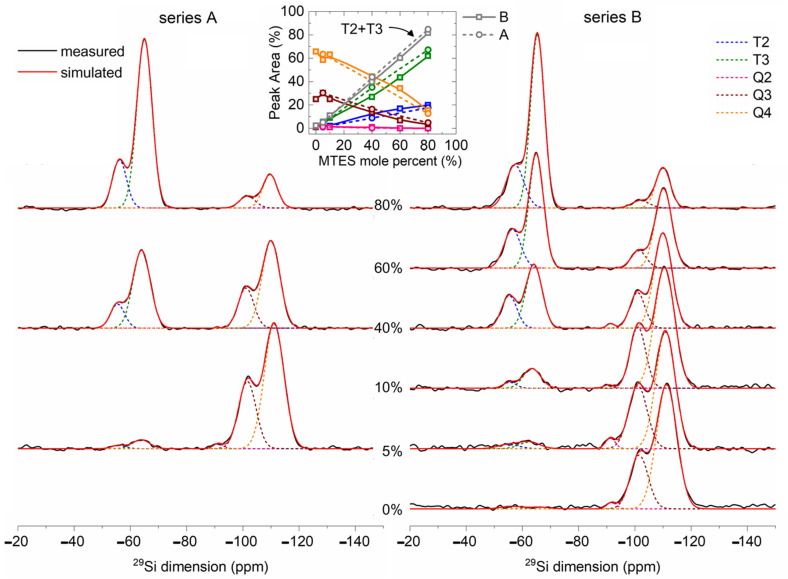
Solid-state ^29^Si-NMR spectra of the sol–gel derived silica from different TEOS/MTES compositions (series A and series B labels represent the pH = 2.0 and pH = 4.5).

**Figure 10 ijms-22-09197-f010:**
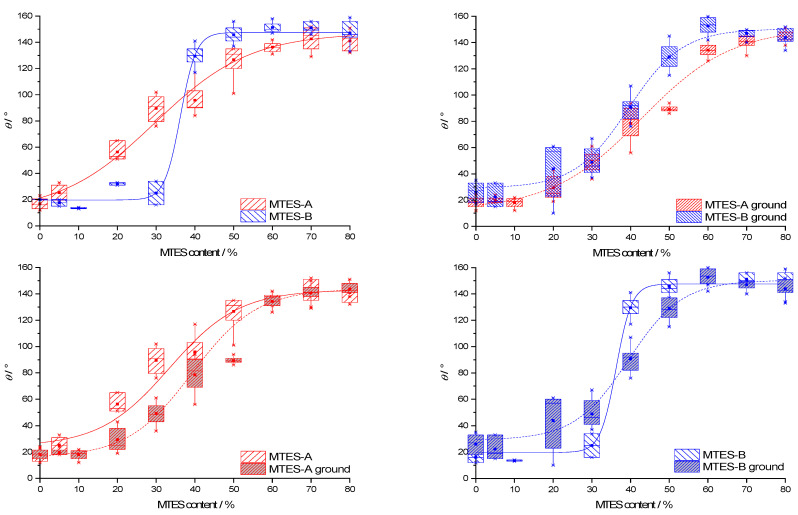
Box plot statistical representation of the advancing contact angles of MTES samples, series A and B, both as-received and ground, in function of the methyl content, grouped in pairs in order to facilitate comparison (series A and series B labels represent the pH = 2.0 and pH = 4.5).

**Figure 11 ijms-22-09197-f011:**
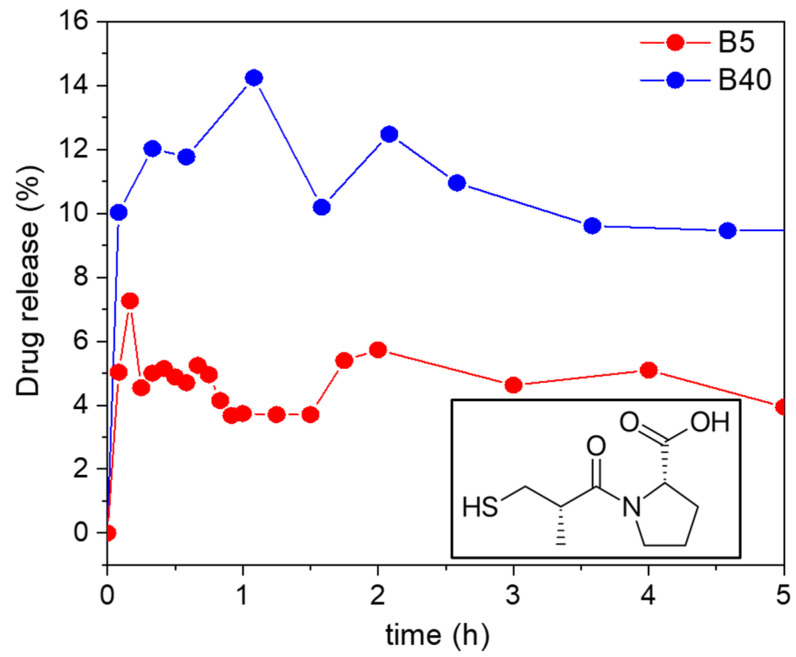
The percent of Captopril (molecular structure in corner) release in time for samples B5 and B40, loaded by adsorption, in hydrochloric acid solution, pH 1.2.

**Figure 12 ijms-22-09197-f012:**
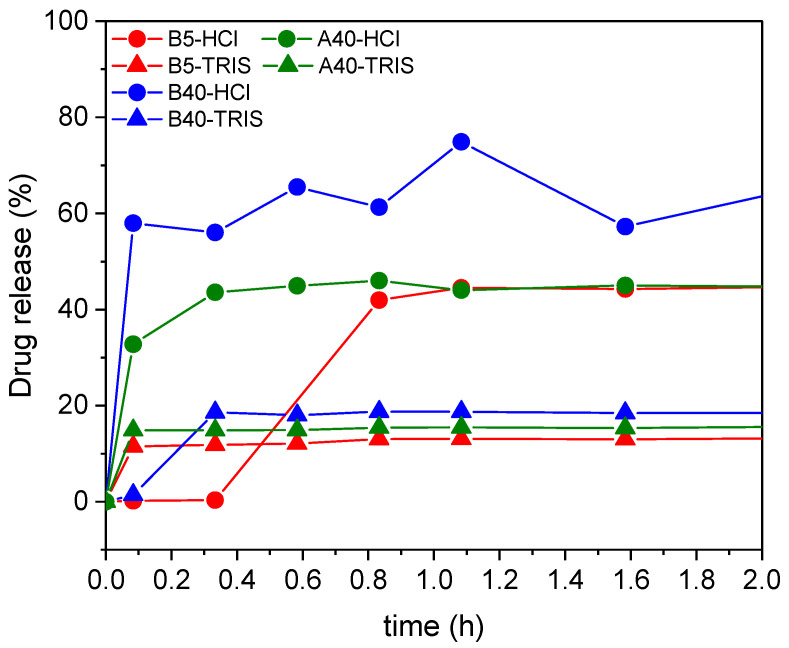
The percent of drug release in time for samples loaded in situ.

**Table 1 ijms-22-09197-t001:** Comparison of the number of species determined from the ^29^Si-MAS NMR spectra for series A and B (series A and series B labels represent the pH = 2.0 and pH = 4.5).

Sample	^29^Si-MAS-NMR										
	Series	Q^4^ (%)	Q^3^ (%)	Q^2^ (%)	T^3^ (%)	T^2^ (%)	T^1^ (%)	Q^4^ + T^3^ (%)	Q^4^ + T^3^ + Q^3^ (%)	Q^3^+Q^2^ + T^2^ (%)	Q^2^ + T^2^ (%)
5	A	60.62	33.37	0.42	3.79	1.79	-	64.41	97.78	35.58	2.21
	B	63.34	32.33	1.51	2.82	0	-	66.16	98.49	33.84	1.51
40	A	40.99	15.32	0	34.87	8.82	-	75.86	91.18	24.14	8.82
	B	43.57	14.75	0.62	26.54	14.51	-	70.11	84.86	29.88	15.13
80	A	12.54	4.65	0	64.28	18.53	-	76.82	81.47	23.18	18.53
	B	17.43	1.58	0	61.18	18.89	0.91	78.61	80.19	20.47	18.89

**Table 2 ijms-22-09197-t002:** Thermal behavior of series B silica xerogels synthesized with different MTES content (series B represents the pH = 4.5).

MTES Content (Mole %)	Mass Loss (%) at			Total Mass Loss (%)
	25–250 °C	250–650 °C	650–800 °C	
0	2.55	0.28	0.022	2.84
5	6.85	4.83	1.08	12.29
20	5.97	5.98	1.19	12.64
40	3.13	7.97	1.18	11.90
60	1.93	10.36	0.90	12.88
80	1.44	11.44	0.72	13.34

**Table 3 ijms-22-09197-t003:** The % of drug release in different buffer solutions.

Sample	Time of Release (h)	% of Captopril Release	
		pH = 1.2	pH = 7.4
Capt-insitu-B5	6 h	43%	13%
	96 h	49.17%	10%
Capt-insitu-B40	7 h	40.16%	16.6%
	24 h	47%	18.4%
Capt-insitu-A40	7 h	43%	15%
	24 h	46.16%	16.62%

## Data Availability

The data presented in this study are available on request from the corresponding author.

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
