# Peer review of "Physicochemical Characterization and Drug Release Properties of Methyl-Substituted Silica Xerogels Made Using Sol–Gel Process"

_ijms, 2021, doi:10.3390/ijms22179197_

Round 1
Reviewer 1 Report
The evaluated paper concerns examination of the influence of the pH and the effect of grafting of organic moieties (methyl groups) on the surface of silica materials.
The silica xerogels were obtained by sol–gel process and characterized by various methods in terms of composition (FTIR, XRD), texture and morphological properties (nitrogen sorption, electron microscopy), hydrophobicity (contact angle), and thermal stability (thermal analysis).
I have noted important data from 29Si CP MAS NMR investigations.
Final remark: minor revision required
Questions:
There was no significant publication in the field of: the synthesis of silicas of the same type, including the sol-gel process, as well as surface modification/treatment.
Both in Figure 4 and in the results description., isotherms are incorrectly called because the curves reflect adsorption and desorption. Therefore, they should be called adsorption/desorption isotherms and the technique of low-temperature nitrogen sorption. The title of this chapter needs also revision.
Please change the specific surface area to surface area as commonly accepted version.
It will be also interesting to see changes in the silanols signals on FTIR spectra.
Please apply UV-Vis instead of UV-vis
The authors should provide reasonable evidence of the degree of methylation of silica due to the number of silanol groups. It is well known that only chemical interactions (condensation) produce stable bonds. Hence, the optimization of modifications seems crucial.
Author Response
Manuscript ID: ijms-1347516
TITLE: Physicochemical characterization and drug release properties of methyl-substituted silica xerogels made by sol–gel process
List of responses to the comments
Reviewer #1:
The evaluated paper concerns examination of the influence of the pH and the effect of grafting of organic moieties (methyl groups) on the surface of silica materials.
The silica xerogels were obtained by sol–gel process and characterized by various methods in terms of composition (FTIR, XRD), texture and morphological properties (nitrogen sorption, electron microscopy), hydrophobicity (contact angle), and thermal stability (thermal analysis).
I have noted important data from 29Si CP MAS NMR investigations.
Final remark: minor revision required
We thank the Reviewer #1 for his/her very positive judgment and for the constructive review of our paper. We revised the manuscript taking into account all his/her suggestions, as reported in the following. All the performed changes are highlighted in yellow throughout the manuscript.
Questions:
There was no significant publication in the field of: the synthesis of silicas of the same type, including the sol-gel process, as well as surface modification/treatment.
Thank you for highlighting our research.
Both in Figure 4 and in the results description., isotherms are incorrectly called because the curves reflect adsorption and desorption. Therefore, they should be called adsorption/desorption isotherms and the technique of low-temperature nitrogen sorption. The title of this chapter needs also revision.
We agree with the Reviewer. According to his/her suggestion, the term “isotherms” has been replaced with “adsorption/desorption isotherms” both in the caption of Figure 2 (we think the Reviewer meant Figure 2 and not Figure 4) and in the results description. The name of the technique was changed according to the Reviewer suggestion, and the title of the sub-section was modified accordingly. See lines 152-155 of the revised version of the manuscript.
Please change the specific surface area to surface area as commonly accepted version.
We thank the Reviewer for his/her observations. Following his/her suggestion, the term “specific surface area” has been replaced with “apparent surface area” throughout the manuscript, nowadays more accepted.
It will be also interesting to see changes in the silanols signals on FTIR spectra.
A sentence was introduced regarding the symmetric Si-OH signal changing, see lines 276-278
Please apply UV-Vis instead of UV-vis
The change has been made throughout all the manuscript.
The authors should provide reasonable evidence of the degree of methylation of silica due to the number of silanol groups. It is well known that only chemical interactions (condensation) produce stable bonds. Hence, the optimization of modifications seems crucial.
According to the authors, from the 29Si-NMR, the sum of the Tx species shows the degree of methylation. The -OH groups containing silicon sites were calculated and presented in the manuscript, see lines 345-351 of the revised version of the manuscript.
Reviewer 2 Report
This work is interesting and quiet good, however, data presentation in the abstract should be added as well the experimental results.
Keywords are not shown in the abstract and should be changed
It need to add more linking words between paragraphs, such as in line 83. This paragraph does not have any correlation between others.
Result were well explained well and detail
It is suggested that if there are several figures shown in a figure, it is better to put (a), (b), (c), etc in each graph in order to separate the explanation between figures (example: fig. 11)
line 583 - hydrophilicity of silica profile are changed become less hydrophilic
Need more explanation for the result, also, need to cite more articles related to this work.
Please re write a good and clear sentences grammatically.
Author Response
Dear Dr. Dudás,
Sorry about the last review report. Please accept our apology and revise
the manuscript based on the previous three reviews.
Have a good day and look forward to receiving your revised version.
Kind regards,
Ms.Ziva Wang,
Assistant Editor
E-Mail: ziva.wang@mdpi.com
Skype: live:.cid.6afd08330a2c60a
IJMS (International Journal of Molecular Sciences)
Impact Factor 5.923 (2021 Journal Citation Reports®)
Homepage: https://www.mdpi.com/journal/ijms
LinkedIn: http://www.linkedin.com/in/ijmsjournal
IJMS channel on Twitter (@IJMS_MDPI)
Reviewer 3 Report
The authors have optimized the synthesis of methyl-substituted silica xerogels made by sol–gel process and they have used it as nanocarrier. The synthesis part is very strong and very interesting. However, the application part (drug release) seems does nor contain sufficient data. My comments are as below:
-In Introduction, the rational behind why pH>4.5 has not been chosen should be explained for readers although it is obvious for experts. It will help junior readers to understand the reaction better.
-Introduction can be enriched by citing different nanocarriers and discuss how silica-based nanocarriers are better.
-A scheme for reaction mechanism can help readers to digest what is happening at different pH.
-Figures 1-3 should be combined.
-The images for contact angle test are better to be presented.
-The loading pH and release pH should be indicated clearly. I cannot get why about 50% of drug cannot be released even after 96 hr at pH=1.2. To me, this nanocarrier is useless. Could you explain?
-Release profile should be investigated by mathematical models. The mechanism of release is unclear and should be clearly discussed.
-MTT should be performed to evaluate the cytotoxicity of the nanocarrier on normal and cancer cells.
Author Response
Manuscript ID: ijms-1347516
TITLE: Physicochemical characterization and drug release properties of methyl-substituted silica xerogels made by sol–gel process
List of responses to the comments
Reviewer #3:
The authors have optimized the synthesis of methyl-substituted silica xerogels made by sol–gel process and they have used it as nanocarrier. The synthesis part is very strong and very interesting. However, the application part (drug release) seems does nor contain sufficient data. My comments are as below:
We are grateful to the Reviewer #3 for the generally positive judgment. The paper has been improved taking into account all the suggestions of the Reviewer. All the performed changes are highlighted in yellow throughout the manuscript.
-In Introduction, the rational behind why pH>4.5 has not been chosen should be explained for readers although it is obvious for experts. It will help junior readers to understand the reaction better.
The following sentence was introduced, see lines 134-135 of the revised version of the manuscript: “The A and B labels represent the gels obtained at pH = 2.0 and pH = 4.5, respectively. These two pH values were chosen taking in account the pH value where the point of zero charge is found and the lowest condensation rate is taking place.” The rest of the cases are explained in lines 70-77: “The use of acid or base catalyzed sol–gel routes modifies the primary particle formation pathway implying different particle shapes. The hydrolysis achieves the lowest reaction rate at pH = 7. For example at pH < 7, it is well-known that hydrolysis and condensation occur successively, the polycondensation process mechanism consists of cluster–cluster aggregation, and primary particles are represented by linear or weakly branched polymer chains. However, under basic conditions (pH > 7), hydrolysis and condensation occur simultaneously, the silica polycondensation follows cluster–particle mechanism, and primary silica particles look like highly branched clusters.”
-Introduction can be enriched by citing different nanocarriers and discuss how silica-based nanocarriers are better.
A paragraph was introduced comparing various carriers with the silica carrier, highlighting the advantage of using the mesoporous silica. See lines: 112-119 of the revised version of the manuscript.
-A scheme for reaction mechanism can help readers to digest what is happening at different pH. -
Without extensive kinetic and spectroscopic measurements, postulating a mechanistic model would only be speculation. Therefore, we do not wish to comment on the hydrolysis and condensation kinetics, as it is out of the scope of the present study. There are several mechanistic investigations in the literature that explain why the rate of the hydrolysis and the steps of the condensation reaction change dramatically from pH=2 to pH=4.5, which in turn alters the morphology of the nanostructured silica backbone.
-Figures 1-3 should be combined.
According to the suggestion of the Reviewer, Figures 2 and 3 were combined to Figure 2a and 2b in the revised version of the manuscript. All the remaining Figures were re-numbered accordingly.
-The images for contact angle test are better to be presented.
Representative contact angle CA images, one for each sample type, series A, respectively B, both as-received and also ground, are presented in the Appendix as Figure A.5.
-The loading pH and release pH should be indicated clearly. I cannot get why about 50% of drug cannot be released even after 96 hr at pH=1.2. To me, this nanocarrier is useless. Could you explain?
An in-depth mechanistic explanation for the retention of drugs are given in the original manuscript lines 434-444, with citing appropriate references. The particular case of the retention in pH=1.2 HCl is highlighted and discussed in details in the original manuscript, lines 482-500. Carriers such as these are very useful for solubilizing drugs of limited hydrophilicity, and delivering them, first in a burst like manner reaching a high concentration, and later maintaining it from the reservoir of the carrier. As the drug is absorbed in the body, the dynamic equilibrium nature of the release process replenishes it and helps maintaining a constant concentration. This is very useful e.g. in the oral administration of drugs for the treatment of pain, acute infections or inflammations.
-Release profile should be investigated by mathematical models. The mechanism of release is unclear and should be clearly discussed.
Drug release – as highlighted multiple times in the manuscript – is burst-like. The mechanism of burst release is the prompt dissolution of the drug from the surface and the open and accessible pores of the carrier particles. Such drug release is not governed by swelling, erosion or limited mass transport (hindered diffusion), and therefore the mathematical models developed for describing these processes are not adequate for the carriers described in the present paper. Thus, release data was fitted with semi-empirical mathematical models in spite of the fact that these models have very limited physico-chemical meaning. Nevertheless, fitting with the first order kinetic model is described in details in Appendix A.
-MTT should be performed to evaluate the cytotoxicity of the nanocarrier on normal and cancer cells.
It is very important to highlight that the silica-based carriers described in the present paper are porous nanostructured carriers, but not nanoparticles. Therefore, nanoparticle-associated toxicity is excluded. The material of the carriers is porous nanostructured and surface modified silica. These materials have been described several times in the literature to be non-toxic and biocompatible. [10.1039/D1EN00026H and 10.1016/j.addr.2012.05.008]
Reviewer 4 Report
The manuscript written by Len et al. reports a comprehensive physico-chemical investigation of hybrid silica xerogels obtained from tetraethoxysilane (TEOS) and increasing amounts (up to 80% molar percent) of methyltriethoxysilane (MTES), at two pH values (2.0, and 4.5, respectively). Using various characterization methods: FT-IR, XRD, 29SI-NMR, TEM, SEM, SANS, nitrogen adsorption-desorption, TG, the authors clearly demonstrated how the structural, morphological, and textural properties of the sol-gel matrix, particularly the hydrophobicity, can be finely tuned in an apparently simple way, by modifying the sol-gel synthesis parameters (molar ratio and pH). Although such hybrid xerogels were already the subject of several reports, I appreciate that the manuscript presents an original approach and will attract scientific interest. Specifically, I would like to highlight the discussion of the FT-IR spectroscopy results, as being one of the most valuable I have seen in the scientific literature of hybrid xerogels (usually, the authors only assign some major adsorption bands). Based on these reasons, I recommend the manuscript for publication, with some observations and recommendations for minor revisions, as it will follow.
- The nitrogen adsorption-desorption isotherms are presented in Figure 2 only up to 20% methyl substitution, for series B. Why? It would have been interesting to see the isotherms also at higher MTES/TEOS ratios (the graphs can be added in the Supplementary material), since the data are discussed in the main text and presented in Table A1.
- Adding the chemical structure of Captopril somewhere in Figure 12 would be useful, to easier follow the discussion concerning the possible interactions with the matrix. In the title of the section 2.10 (page 14, line 39), please replace "Captorpil" with "Captopril". I think that Captopril should be capitalized throughout the text. In the legend of Figure 12, delete " "(Milyen bufferbe?)", and add "in hydrochloric acid solution, pH 1.2".
- The controlled release study of Captopril is less elaborated compared to the characterization of the matrix but can be considered a first tentative to correlate the physico-chemical characteristics of these materials with the drug retention and release properties. However, in the Introduction part some other studies concerning the controlled release of Captopril should be cited, particularly using porous silica matrices (like as https://doi.org/10.1016/j.micromeso.2005.12.004). Since Captopril release was not optimal, possibilities to improve it or maybe to test other drugs better fitted with the structure of these xerogels should be also mentioned.
- The loading of Captopril in both versions, by adsorption and by entrapment, should be also given (as mg per g support).
Author Response
Manuscript ID: ijms-1347516
TITLE: Physicochemical characterization and drug release properties of methyl-substituted silica xerogels made by sol–gel process
Reviewer #4:
The manuscript written by Len et al. reports a comprehensive physico-chemical investigation of hybrid silica xerogels obtained from tetraethoxysilane (TEOS) and increasing amounts (up to 80% molar percent) of methyltriethoxysilane (MTES), at two pH values (2.0, and 4.5, respectively). Using various characterization methods: FT-IR, XRD, 29SI-NMR, TEM, SEM, SANS, nitrogen adsorption-desorption, TG, the authors clearly demonstrated how the structural, morphological, and textural properties of the sol-gel matrix, particularly the hydrophobicity, can be finely tuned in an apparently simple way, by modifying the sol-gel synthesis parameters (molar ratio and pH). Although such hybrid xerogels were already the subject of several reports, I appreciate that the manuscript presents an original approach and will attract scientific interest. Specifically, I would like to highlight the discussion of the FT-IR spectroscopy results, as being one of the most valuable I have seen in the scientific literature of hybrid xerogels (usually, the authors only assign some major adsorption bands). Based on these reasons, I recommend the manuscript for publication, with some observations and recommendations for minor revisions, as it will follow.
We thank to the Reviewer #4 for the positive judgment and the helpful comments. The paper has been improved taking into account all the suggestions of the Reviewer. All the performed changes are highlighted in yellow throughout the manuscript.
- The nitrogen adsorption-desorption isotherms are presented in Figure 2 only up to 20% methyl substitution, for series B. Why? It would have been interesting to see the isotherms also at higher MTES/TEOS ratios (the graphs can be added in the Supplementary material), since the data are discussed in the main text and presented in Table A1.
Starting from 20% methyl substitution, the type of the hysteresis is not changing. For a better visibility, we chose to present only the most characteristic isotherms. A Figure has been added to the Supplementary materials with the rest of the isotherms.
- Adding the chemical structure of Captopril somewhere in Figure 12 would be useful, to easier follow the discussion concerning the possible interactions with the matrix. In the title of the section 2.10 (page 14, line 39), please replace "Captorpil" with "Captopril". I think that Captopril should be capitalized throughout the text. In the legend of Figure 12, delete, and add "in hydrochloric acid solution, pH 1.2".
According to the suggestion of the Reviewer, the chemical structure of Captopril was added in Figure 11 (Figure 12 in the original version of the manuscript). All the other suggested changes have been made and highlighted in the manuscript.
- The controlled release study of Captopril is less elaborated compared to the characterization of the matrix but can be considered a first tentative to correlate the physico-chemical characteristics of these materials with the drug retention and release properties. However, in the Introduction part some other studies concerning the controlled release of Captopril should be cited, particularly using porous silica matrices (like as https://doi.org/10.1016/j.micromeso.2005.12.004). Since Captopril release was not optimal, possibilities to improve it or maybe to test other drugs better fitted with the structure of these xerogels should be also mentioned.
The controlled release of Captopril from mesoporous silica is now mentioned in the revised Section 2.10. The release of Captopril from MCM-41 carriers in simulated intestine and gastric fluids start with a burst phase in which ca. 60% of the drug dissolves. This is followed by a slow retarded phase reaching ca. 90% cumulative release. The overall rate is significantly slower in simulated intestine fluid due to its near neutral pH. Under these conditions, Captopril becomes deprotonated, which causes its strong interaction with silica that in turn limits the release rate.
As now mentioned in the revised Section 2.11, more suitable drugs that could benefit from loading into the studied xerogels are nonsteroidal anti-inflammatory drugs (e.g. Ibuprofen, Ketoprofen). Higher loading and cumulative release are expected from these drugs due to their partially hydrophobic nature. Their controlled solubilisation by the studied xerogels could be advantageous for their therapeutic action.
- The loading of Captopril in both versions, by adsorption and by entrapment, should be also given (as mg per g support).
The loading of Captopril was of 1086.5 mg captopril/g of silica support. This value can be considered for both, the adsorption and in the case of entrapment too (because in the case of method by entrapment the same amount of drug was used and the same synthesis recipe has been followed with the same precursors quantities, and therefore concluding that the same theoretically amount of silica was obtained).